# Modelling the concentration of anti-SARS-CoV-2 immunoglobulin G in intravenous immunoglobulin product batches

Sara Stinca[1☯], Thomas W. Barnes[1☯], Peter Vogel[2], Wilfried Meyers[2], Johannes Schulte-Pelkum[3], Daniel Filchtinski[3], Laura Steller[3], Thomas Hauser[1], Sandro Manni[1], David F. Gardiner[4], Sharon Popik[4], Nathan J. Roth[1], Patrick Schuetz[1]*

1 Department of Bioanalytical Sciences, Plasma Product Development, Research & Development, CSL Behring AG, Bern, Switzerland, 2 Global Digital Core, Plasma Product Development, Research & Development, CSL Behring Innovation GmbH, Marburg, Germany, 3 Assay Design, Thermo Fisher Scientific ImmunoDiagnostics Phadia GmbH, Freiburg, Germany, 4 Immunology, CSL Behring, King of Prussia, Pennsylvania, United States of America

☯ These authors contributed equally to this work.
* Patrick.Schuetz@cslbehring.com

**Data Availability Statement:** All relevant data are within the paper and its Supporting information files.

## Abstract

### Background

Plasma-derived intravenous immunoglobulin (IVIg) products contain a dynamic spectrum of immunoglobulin (Ig) G reactivities reflective of the donor population from which they are derived. We sought to model the concentration of anti-severe acute respiratory syndrome coronavirus 2 (SARS-CoV-2) IgG which could be expected in future plasma pool and final-product batches of CSL Behring's immunoglobulin product Privigen.

### Study design and methods

Data was extracted from accessible databases, including the incidence of coronavirus disease 2019 and SARS-CoV-2 vaccination status, antibody titre in convalescent and vaccinated groups and antibody half-life. Together, these parameters were used to create an integrated mathematical model that could be used to predict anti-SARS-CoV-2 antibody levels in future IVIg preparations.

### Results

We predict that anti-SARS-CoV-2 IgG concentration will peak in batches produced in mid-October 2021, containing levels in the vicinity of 190-fold that of the mean convalescent (unvaccinated) plasma concentration. An elevated concentration (approximately 35-fold convalescent plasma) is anticipated to be retained in batches produced well into 2022. Measurement of several Privigen batches using the Phadia™ EliA™ SARS-CoV-2-Sp1 IgG binding assay confirmed the early phase of this model.

**Funding:** Funding for this work was provided by CSL Behring AG and Thermo Fisher Scientific ImmunoDiagnostics Phadia GmbH. Editorial assistance was funded by CSL Behring.

**Competing interests:** I have read the journal's policy and the authors of this manuscript have the following competing interests: SS, TB, PV, WM, DG, TH, SM, SP, NR and PS are employees of CSL Behring. JSP, DF and LS are employees of Thermo Fisher Scientific ImmunoDiagnostics Phadia, GmbH.

## Conclusion

The work presented in this paper may have important implications for physicians and patients who use Privigen for indicated diseases.

## Background

Intravenous immunoglobulin (IVIg) products are used as therapeutic agents for several auto-immune, immunodeficiency and infectious diseases [1]. Manufactured from pooled human plasma donations, IVIg products contain the spectrum of immunoglobulin G (IgG) reactivities present in the donor population, which broadly reflects disease incidence and vaccination rates in society. The spectrum and distribution of disease-specific IgG species is dynamic, changing both geographically and temporally with disease prevalence in donor populations [2]. Consequent to the coronavirus disease 2019 (COVID-19) pandemic, there has been a particularly rapid increase in the prevalence of anti-severe acute respiratory syndrome corona-virus 2 (SARS-CoV-2) IgG in the population, arising from both natural infection and vaccination [3, 4]. The level of anti-SARS-CoV-2 IgG in such products may have clinical relevance and this information may be useful for physicians who currently treat patients with immuno-globulin products.

Here, we sought to model the trajectory of the increase in anti-SARS-CoV-2 antibodies in the donor population to predict the levels of anti-SARS-CoV-2 IgG that could be expected in future batches of CSL Behring's IVIg product (Privigen).

## Methods

### Data extraction and grouping of donors by natural infection and vaccination status

Literature and publicly available databases detailing COVID-19 prevalence, vaccination rate, anti-SARS-CoV-2 antibody peak titre and rate of decay (half-life) were interrogated for modelling purposes. Where possible, data was extracted specifically for individuals aged 20–50 years who reside in the USA, since this age group and location best reflects the demographic of CSL Behring's donor population. No other restrictions pertaining to the donor population demographic (e.g., race or gender) were applied. The donor population was divided into six groups representing possible combinations of infection and vaccination status as follows: donors naïve to COVID-19, who had received zero, one or two vaccine doses (groups 1–3), and donors who experienced a natural COVID-19 infection with the same vaccination statuses as above (groups 4–6). Each group was assigned an average anti-SARS-CoV-2 spike antibody concentration (AUC) based on the findings of Krammer *et al.* [3] (S1 Table).

### Model construction and statistical methods

**Modelling temporal changes in relative percentage classified donors ("population model").** The relative proportion of the donor population residing in each group was calculated on a weekly basis, based on data from the US Centres for Disease Control and Prevention (CDC) [5]. The population could be defined by the proportions $p_i(j)$ (percentage of population in group $i$ in calendar week $j$) using the following terms and auxiliary conditions:

$NI_j$ – percentage of natural infections in relevant population (CDC data)

$FV_j$ – percentage of relevant population with first vaccination (CDC data)

$SV_j$ – percentage of relevant population with second vaccination (CDC data)

$\Delta NI_j = NI_j - NI_{j-1}$ – change in natural infection percentage since previous week

$\Delta FV_j = FV_j - FV_{j-1}$ – change in first vaccination percentage since previous week

$\Delta SV_j = SV_j - SV_{j-1}$ – change in second vaccination percentage since previous week

Furthermore, the following auxiliary conditions were taken into account:

$p_{1j} = 100 - \sum_{i=2}^{6} p_i(j)$ for all $j$

$p_{4j} = \Delta NI_j + p_4(j-1)$

$\Delta FV_j$ is proportionally split between groups $i = 2$ and $i = 5$ according to the split between the groups 1 and 4 at the previous timepoint

$\Delta SV_j$ is proportionally split between groups $i = 3$ and $i = 6$ according to the split between the groups 2 and 5 at the previous timepoint

The prediction of future development of the population curves, beginning from July 2021 until March 2022 was based on a logistic curve plus a linear component which eventuates with approximately 10.2% of the donor population having a natural infection and approximately 71.4% being fully vaccinated. Data derived from these calculations are shown in S3 Table.

**Modelling antibody half-life in blood ("decay model").** The decay model calculates the concentration $T_{ik}$ of a donor after $k$ weeks residence time in group $i$. An exponential decay was assumed with a pre-specified half-life after the initial transition phase of about three weeks until the peak titre is reached (S2 Table). The concentration at any given time in the transition phase is provided in S2 Table, and beyond three weeks was calculated as follows:

$T_{ik} = APT_i \cdot e^{-\theta_i \cdot (k-3)}$ with $\theta_i = -\ln(2)/(\frac{HLT_i}{7})$ for.

$APT_i$ – average peak titre in group $i$

$HLT_i$ – half-life of titre in group $i$ in days

**Modelling the duration spent by donors in each group ("residence time model").** The contribution of an individual donor to the plasma pool is determined by their personal concentration at donation. This personal concentration changes over time and relates to the group of the donor (driving the peak concentration) as well as the time since entrance into this group (decay starts after reaching the peak concentration; the change in concentration after vaccination for each group is shown in S1 Table). The exponential function used in the Decay Model leads to a modelled reduction of the personal donor concentration to 0.1% of the peak concentration after ten cycles of the pre-specified half-life. How long a donor stays in a given group must therefore be modelled. Final calculations were performed with a sufficiently high number of weeks as maximum residence time per group. In the residence time model, the population proportion of the group is therefore split into sub-groups ($r_{ijk}$) with respect to their residence time $k$ (percentage of group $i$ in week $j$) within that group.

The following auxiliary conditions were taken into account:

$p_{ij} = \sum_{k=1}^{\infty} r_{ijk}$ for all $i$ and $j$

$r_{ij1}$ – additions to group $i$ in calendar week $j$ since calendar week $j - 1$

$l_{ij}$ – leaves from group $i$ since calendar week $j − 1$ due to natural infection or further vaccination

The $l_{ij}$ were proportionally split based on the $r_{i,j−1,k}$ for $k = 1, . . ., 10$ and $r_{ijk} = r_{i,j−1,k−1} − l_{ijk}$.

**Creation of an integrated model (combined population, residence time and decay models).** Based on the population, decay, and residence time models, the absolute and relative mean convalescent plasma concentration (MCPC) of anti-SARS-CoV-2 IgG in plasma pools were calculated weekly as weighted averages of the individual components:

$$T_j = \sum_{i=1}^{6} \sum_{k=0}^{10} r_{ijk} \cdot T_{ik}$$

The predictions made by the integrated model were then extrapolated to project the anti-SARS-CoV-2 IgG concentration in Privigen: (i) to account for manufacturing lead-times, concentrations were projected forwards by 4.5 months; (ii) absolute concentrations were multiplied by a factor of ten, due to the final formulation of Privigen being 10% IgG (~10-fold the concentration of IgG in plasma); and (iii) absolute concentrations were converted from AUC (as per Krammer *et al.* [3]) to U/mL, the unitage reported using the Phadia™ EliA™ SARS-CoV-2-Sp1 IgG assay (EliA S1-IgG, ThermoFisher Scientific, Uppsala, Sweden). The latter conversion was made by equating the mean convalescent titre with the mean of the results (determined using the EliA S1-IgG assay) for six plasma pools produced from donations made by exclusively convalescent donors as previously reported [3].

Further details of statistical methods and approaches applied in this study can be found in the Supporting information.

## Determination of anti-SARS-CoV-2 antibody concentration in Privigen batches

A total of 49 batches of Privigen were analysed for anti-SARS-CoV-2 IgG concentration using the Phadia™ EliA™ SARS-CoV-2-Sp1 IgG assay (EliA S1-IgG) assay, as previously described [6]. These batches were processed to final product between January 2018 and June 2021, encompassing pre-pandemic, and pre- and post-vaccine rollout time periods. In addition, six separate plasma pools, produced from donations made by exclusively convalescent donors prior to vaccination roll-out were analysed using the same technique. IRB/ethical committee approval was not required. All plasma donors signed the CSL Behring general consent form for use of their plasma in research.

## Results & discussion

### Modelling predicts the anti-SARS-CoV-2 antibody level in donor plasma and future Privigen batches

We constructed a mathematical model, drawing upon the CDC COVID Data Tracker [5] for COVID-19 incidence and vaccination rates for the 20–50 year old age group in the USA. Several studies have reported a substantial increase (10–100-fold) in peak antibody titre as a consequence of vaccination, in comparison to natural infection [3–5, 7, 8]. This information was considered to support the predictive ability of our model, as the peak appears approximately three weeks following vaccination [3]. The half-life of anti-SARS-CoV-2 antibodies in blood was also an important input parameter for our model. Various reports have calculated this to be between 20.4 and 46.9 days in convalescent donors [9, 10], and between 52 days and 65 days in vaccinated individuals [11, 12]. These parameters were combined to generate an

integrated model with predictive capability for future concentrations of anti-SARS-CoV-2 IgG in Privigen batches.

Recently, Krammer and co-workers demonstrated that individuals who had previously suffered from SARS-CoV-2 infection had significantly elevated antibody levels following vaccination in comparison to those who were naïve for the disease [3]. Drawing on this study, which was based on measurements made using a Spike-1 protein binding ELISA and reported in the unitage of Area Under Curve (AUC), multiple groups were created for the purposes of modelling, and median anti-SARS-CoV-2 IgG concentrations were assigned to each. These groups comprised: COVID-19 naïve donors who received zero, one or two vaccine doses (Groups 1–3), and previously infected donors with the same vaccination status as above (Groups 4–6) (S1 Table). The population and titre sub-models, which feed the overall mathematical model, reflect the differences between the six groups and demonstrate meaningful predictions of the anti-SARS-CoV-2 antibody levels in the target population.

The population model specifies for each group $i$ and timepoint $j$ (in weeks) the proportion $p_{i,j}$ of donors in that group and estimates these proportions from publicly available data. In addition, it allows the derivation of transitions of donors into and out of respective groups as a result of an infection or vaccination event, and furthermore to deduce the proportion $p_{i,j,k}$ of donors in group $i$ at time point $j$ that transitioned $k$ weeks prior into this group and still belong to the group (residence time of $k$ weeks). Given the strong effect of transitions on the initial titer and its decay, these proportions $p_{i,j,k}$ are key to the prediction of the temporal evolution of the anti-SARS-CoV-2 IgG titre. The mathematical details of the population and the titre model are provided in the Supporting information. As shown in S1 Table, anti-SARS-CoV-2 titre increases strongly in the first two to three weeks after an infection or vaccination event before the exponential decay starts to dominate the titre evolution. For the modelling of the titre in each group, we therefore differentiate between the transition phase of the first three weeks that relies on the titre values displayed in S1 Table, and the exponential decay of the titre (S2 Table). In combination with the knowledge about the proportions $p_{i,j,k}$, we obtain the predictions for the temporal evolution of the anti-SARS-CoV-2 titre. Fig 1 displays the observed temporal evolution of the proportions of the donor groups until mid-August 2021, as well as their predicted evolution until end of March 2022.

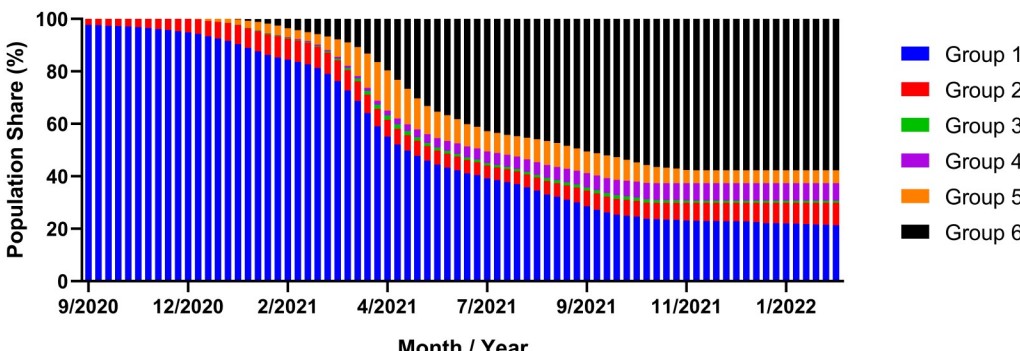

**Fig 1. Proportions of donor groups and their predicted evolution over time.** Groups 1–3 represent SARS-CoV-2 infection-naïve individuals who had received zero, one or two vaccine doses, respectively. Groups 4–6 represent previously infected individuals who had received zero, one or two vaccine doses, respectively. Actual proportions of donors were calculated up to August 2021, with predicted values estimated from September 2021 until March 2022.

## Extrapolating the SARS-Cov-2 IgG concentration prediction model to Privigen

The anti-SARS-CoV-2 IgG concentration predictions made by the integrated model for donor plasma were then extrapolated to the IVIg product, taking into account manufacturing parameters, including lead-time between plasma collection and final-product output (4.5 months) and final product IgG concentration (approximately 10-fold that of plasma).

Furthermore, we adjusted the output of the model to account for the availability of analytical methods for sample analysis. Currently, no standardised analytical method for assessing anti-SARS-CoV-2 antibody concentration exists [12], making the selection of a read-out for the model non-trivial. At our disposal was the Phadia™ EliA™ SARS-CoV-2-Sp1 IgG assay (EliA S1-IgG) which has a reported unitage of Units per millilitre (U/mL), where one U/mL corresponds to four WHO International Units per millilitre. This method is CE approved and received FDA EUA approval in January 2021. Furthermore, a strong correlation to cell-based live-virus SARS-CoV-2 neutralisation assays has been previously observed [6].

We measured six convalescent plasma pools, each comprising approximately 250 donations made by non-vaccinated donors (essentially Group 4), using the EliA S1-IgG method and observed a mean convalescent plasma concentration (MCPC) of 81 U/mL (S4 Table). In comparison, Krammer *et al.* measured a value of 90 AUC for the corresponding non-vaccinated, previously infected cohort [3]. We therefore multiplied the read-outs of the model by a factor of 0.9 to provide values in EliA S1-IgG unitage and allow for subsequent measurements of Privigen batches to be compared to the model predictions.

The model predicted that concentrations of anti-SARS-CoV-2 IgG in Privigen would surpass the MCPC of 81 U/mL in April 2021, with a peak concentration (~15,400 U/mL; 190-fold of the MCPC) anticipated in mid-October 2021 (Fig 2A).

Due to antibody decay, levels are predicted to decline from this time over the course of several months. This is punctuated by a short levelling period at the beginning of 2022 (~11,000 U/mL; 135-fold of the MCPC), as a consequence of a predicted increase in the proportion of double-vaccinated donors in September 2021. Nevertheless, the predicted concentration remains well above the MCPC (~30-fold) for the duration of the model.

## Measurement of actual anti-SARS-CoV-2 IgG concentration in Privigen samples

Alongside the mathematical model, we also tested 49 Privigen batches collected prior to, and during, the COVID-19 pandemic, using a quantitative Spike-1-IgG-binding method (EliA S1-IgG). As expected, concentrations remained at baseline for batches manufactured up until November 2020 (Fig 2B). However, in February 2021, there was a consistent rise in concentration that continued up until August 2021. The maximum concentration that was observed was 824 U/mL (10.2-fold the MCPC). This data is in line with the predictions made by the mathematical model, providing validation of the findings of the early phases of the model.

## Discussion

Using modelling, partially validated by early phase data in Privigen batches, this study demonstrates that Privigen contains anti-SARS-CoV-2 IgG at levels well above the mean levels of convalescent plasma from unvaccinated individuals, and suggests that these levels may be maintained long-term, which may have several potential clinical implications. Firstly, the success of several COVID-19 vaccines in preventing COVID-19 infection [13], coupled with the substantially elevated anti-SARS-CoV-2 antibody levels of vaccinated individuals [4, 7, 8],

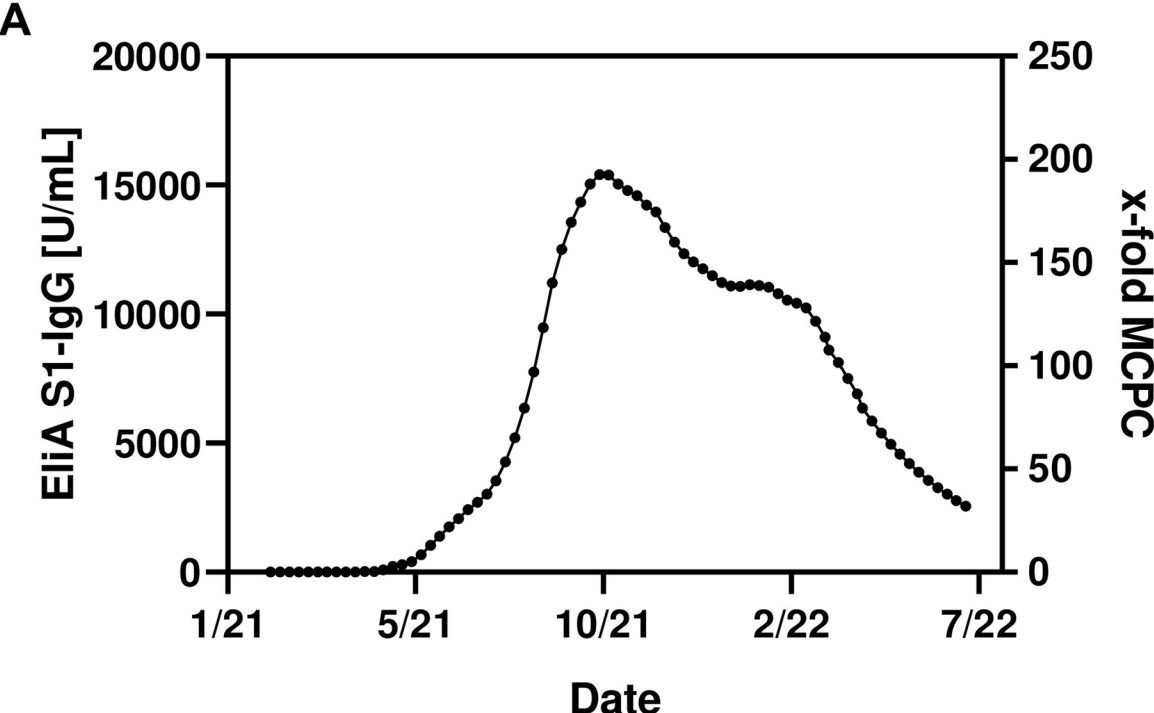

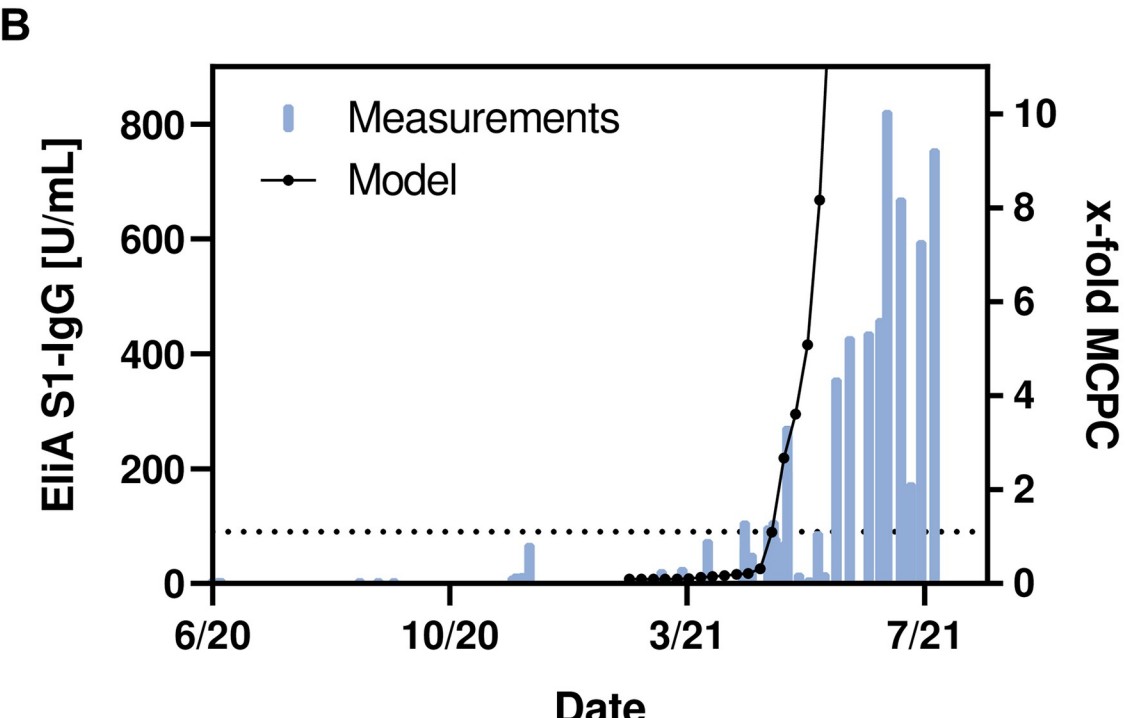

**Fig 2. Modelled and measured levels of anti-SARS-CoV-2 IgG in Privigen over time. (A)** Modelled concentrations of anti-SARS-CoV-2 IgG in Privigen batches from January 2021 until August 2022 and **(B)** modelled (black points) and measured concentrations (blue bars) of anti-SARS-CoV-2 IgG in Privigen batches over time from June 2020 until July 2021. Both absolute (as measured using EliA S1-IgG [left y-axis]), and relative to mean convalescent plasma concentration (MCPC) (right y-axis) are shown. The dotted line in part B indicates a concentration of 81 U/mL, equivalent to one-fold the MCPC.

suggests that administration of IVIg containing a high concentration of COVID-19-specific IgG may provide for the transfer of anti-SARS-CoV-2 IgG. While studies investigating the use of convalescent plasma in the treatment of COVID-19 have not shown consistent clinical benefit, these results may reflect heterogeneity of study design or administration too late in the course of disease [13–15]. Furthermore, the use of high-titre plasma from vaccinated individuals has not yet been tested for clinical utility. Based on our model, we predict the concentration of anti-SARS-CoV-2 IgG in Privigen to be 190-fold higher than that of convalescent plasma, which suggests that such high levels could thus be present in the recipient after administration.

It has previously been shown that vaccination induces distinct humoral RBD-specific IgG responses in vaccinated and non-vaccinated subjects [16]. Naïve individuals receiving their first dose of mRNA vaccine developed a SARS-CoV-2 specific antibody response with a subclass profile similar to that induced by a natural infection and increased concentrations of all IgG subclasses simultaneously also upon re-vaccination. Subjects with pre-existing immunity also showed an increase in all RBD-specific IgG subclasses after initial mRNA vaccination. However, a recent study profiled vaccine-induced polyclonal and monoclonal antibodies and found that the polyclonal immune response to mRNA vaccination exceeds titres seen in convalescent subjects but are characterised by a high ratio of non-neutralising antibodies [17]. Calculations of spike binding to neutralisation titres showed the highest proportion of binding to neutralisation in vaccinated rather than convalescent subjects, indicating the generation of less neutralising antibodies upon mRNA vaccination. The Privigen process is capable of purifying IgGs with broad donor specificities including the physiological subclass distribution. It remains a topic of future investigation to further characterise SARS-CoV-2 specific antibodies in IVIg preparations. Furthermore, the clinical implication of administration of IVIg containing antibody repertoires for anti-SARS-CoV-2 reactivities has not been tested and needs to be further evaluated.

Inadequate immune response following vaccination in patients receiving IVIg products has been previously described, particularly for the live-attenuated mumps-measles-rubella vaccine [18] and, less significantly, for protein-based vaccination [19]. Given the variety of COVID-19 vaccines in development (mRNA, live-attenuated and protein), the magnitude of any clinical impact on vaccine response may vary [20]. Other factors may include the timing of Ig administration relative to the vaccine as well as circulating viral variants in the population at the time of patient exposure. Until further data is available, clinicians should exclusively consider authorised vaccines for the prevention of SARS-CoV-2 disease and approved treatment options for patients suffering from COVID-19. It is important to note that Privigen is not currently approved for treatment of COVID-19; our study is therefore intended purely for the guidance of physicians currently prescribing this product for indicated diseases. Indeed, during the on-going pandemic, questions regarding the current and future levels of anti-SARS-CoV-2 antibodies in CSL Behring's immunoglobulin products have constituted approximately 65% and 45% of patient and health-care provider enquiries to our organisation, respectively. With this in mind, it is vital to convey the results of this study to the wider medical community.

Due to current uncertainty as to the requirement and timing of third vaccine doses, we chose to incorporate only the two-dose strategy into the model. Nevertheless, inclusion of a third dose component would have a significant impact on the dynamics of the groups described, leading to an increase in both the expected magnitude and longevity of anti-SARS-Cov-2 IgG in Privigen batches. Furthermore, the model is not designed to be robust against the possible emergence of novel SARS-CoV-2 strains able to elicit immune memory responses. Developments in these two uncertain aspects of the COVID-19 pandemic, and their incorporation into the model described here, will be the subject of future work.

## Conclusions

According to our model, anti-SARS-CoV-2 IgG concentration in Privigen batches is likely to peak in mid-October 2021, with levels of anti-SARS-CoV-2 IgG estimated to be at approximately 190-fold that of the mean convalescent (unvaccinated) plasma concentration. Though a decline in titre is expected, Privigen batches are anticipated to retain anti-SARS-CoV-2 IgG levels well above that of convalescent plasma into 2022, which may have implications for physicians using immunoglobulin therapy in the clinic.

## Supporting information

**S1 Table. Mean anti-SARS-CoV-2 spike antibody concentration for each donor group, derived from Krammer et al. [3].**
(DOCX)

**S2 Table. Anti-SARS-CoV-2 half-lives utilised for each donor group.**
(DOCX)

**S3 Table. Relative proportion (in percent) of donors in each group per week (derived from data obtained from US CDC).**
(DOCX)

**S4 Table. Observed EliA S1-IgG concentration for convalescent plasma pools (250 donations per pool).**
(DOCX)

**S5 Table. Privigen batches used in this study.**
(DOCX)

**S1 File.**
(DOCX)

**S1 Data.**
(XLSX)

## Acknowledgments

The authors would like to acknowledge the following for logistical support, laboratory analysis, and statistical input: Jasmin Fischer, Dominique Haroutel, Claudia Hadorn and Maya Shevlyakova. Editorial assistance was provided Meridian HealthComms Ltd.

## Author Contributions

**Conceptualization:** Nathan J. Roth, Patrick Schuetz.

**Data curation:** Sara Stinca, Thomas W. Barnes, Johannes Schulte-Pelkum.

**Formal analysis:** Sara Stinca, Thomas W. Barnes, Peter Vogel, Wilfried Meyers, Johannes Schulte-Pelkum.

**Investigation:** Peter Vogel, Johannes Schulte-Pelkum, Daniel Filchtinski, Laura Steller, Thomas Hauser.

**Methodology:** Sara Stinca, Thomas W. Barnes, Peter Vogel, Wilfried Meyers, Johannes Schulte-Pelkum, Daniel Filchtinski, Laura Steller, Thomas Hauser.

**Project administration:** Sara Stinca, Thomas W. Barnes.

**Resources:** Thomas W. Barnes, Nathan J. Roth, Patrick Schuetz.

**Software:** Peter Vogel.

**Supervision:** Thomas W. Barnes, Nathan J. Roth, Patrick Schuetz.

**Visualization:** Sara Stinca, Thomas W. Barnes, Peter Vogel, Thomas Hauser.

**Writing – original draft:** Thomas W. Barnes, Peter Vogel, David F. Gardiner.

**Writing – review & editing:** Thomas W. Barnes, Peter Vogel, Wilfried Meyers, Sandro Manni, Sharon Popik.

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
