## [Decision Letter · Decision Letter 0]

26 Oct 2021

Modelling the concentration of anti-SARS-CoV-2 immunoglobulin G in intravenous immunoglobulin product batches

PONE-D-21-30527

Dear Dr. Schuetz,

We’re pleased to inform you that your manuscript has been judged scientifically suitable for publication and will be formally accepted for publication once it meets all outstanding technical requirements.

Kind regards,

Etsuro Ito

Academic Editor

PLOS ONE

Reviewers' comments:

Reviewer's Responses to Questions

**Comments to the Author**

1. Is the manuscript technically sound, and do the data support the conclusions?

Reviewer #1: Yes

Reviewer #2: Yes

2. Has the statistical analysis been performed appropriately and rigorously? 

Reviewer #1: Yes

Reviewer #2: I Don't Know

3. Have the authors made all data underlying the findings in their manuscript fully available?

Reviewer #1: Yes

Reviewer #2: Yes

4. Is the manuscript presented in an intelligible fashion and written in standard English?

Reviewer #1: Yes

Reviewer #2: Yes

5. Review Comments to the Author

Reviewer #1: This is a rigorous modelling analysis, complemented by empirical data on SARS-CoV-2 Ab levels, in IVIG produced by one major plasma fractionation company over the previous several years with projections into the future as vaccinations and infections evolve and Abs peak and subsequently wane. The investigators predict the concentration of anti-SARS-CoV-2 IgG in Privigen to be 190-fold higher than that of convalescent plasma, which suggests that such high levels could thus be present in the recipient after administration. The paper is very well developed and written with appropriate caveats to the model findings and implications of the projections for prevention of SARS-CoV-2 infection and treatment of COVID-19 disease.

Reviewer #2: This is an interesting contribution which will be useful for physicians, patients and patient organizations. However, as it it consists of predictive modeling, will be replaced by empirical data. This is recognized and stated by the authors. With those limitations, it may be published, but can not be considered definitive.

6. PLOS authors have the option to publish the peer review history of their article (what does this mean?). If published, this will include your full peer review and any attached files.

Reviewer #1: **Yes: **Michael Busch, MD, PhD

Reviewer #2: No

---

## [Editor Report · Acceptance letter]

17 Nov 2021

PONE-D-21-30527 

Modelling the concentration of anti-SARS-CoV-2 immunoglobulin G in intravenous immunoglobulin product batches 

Dear Dr. Schuetz:

I'm pleased to inform you that your manuscript has been deemed suitable for publication in PLOS ONE. Congratulations! Your manuscript is now with our production department. 

Kind regards, 

on behalf of

Prof. Etsuro Ito 

Academic Editor

PLOS ONE